# Recurrent Coevolutionary Feature Embedding Processes for Recommendation

**Hanjun Dai,*  Yichen Wang,*  Rakshit Trivedi & Le Song**
College of Computing
Georgia Institute of Technology
Atlanta, GA 30332, USA
{hanjundai,yichen.wang,rstrivedi}@gatech.edu, lsong@cc.gatech.edu

## Abstract

Recommender systems often use latent features to explain the behaviors of users and capture the properties of items. As users interact with different items over time, user and item features can influence each other, evolve and co-evolve over time. To accurately capture the fine grained *nonlinear* coevolution of these features, we propose a recurrent coevolutionary feature embedding process model, which combines recurrent neural network (RNN) with a multi-dimensional point process model. The RNN learns a nonlinear representation of user and item embeddings which take into account mutual influence between user and item features, and the feature evolution over time. We also develop an efficient stochastic gradient algorithm for learning parameters. Experiments on diverse real-world datasets demonstrate significant improvements in user behavior prediction compared to state-of-the-arts.

## 1 Introduction

E-commerce platforms and social service websites, such as Reddit, Amazon, and Netflix, attracts thousands of users every second. Effectively recommending the appropriate service items to users is a fundamentally important task for these online services. It can significantly boost the user activities on these sites and leads to increased product purchases and advertisement clicks.

The interactions between users and items play a critical role in driving the evolution of user interests and item features. For example, for music streaming services, a long-time fan of Rock music listens to an interesting Blues one day, and starts to listen to more Blues instead of Rock music. Similarly, a single music may also serve different audiences at different times,*e.g.*, a music initially targeted for an older generation may become popular among the young, and the features of this music need to be updated. Furthermore, as users interact with different items, users' interests and items' features can also *co-evolve* over time, *i.e.*, their features are intertwined and can influence each other:

- *User → item*. In online discussion forums such as Reddit, although a group (item) is initially created for statistics topics, users with very different interest profiles can join this group. Hence, the participants can shape the features of the group through their postings. It is likely that this group can finally become one about deep learning because most users concern about deep learning.
- *Item → user*. As the group is evolving towards topics on deep learning, some users may become more interested in deep learning topics, and they may participate in other specialized groups on deep learning. On the opposite side, some users may gradually gain interests in pure math groups, lose interests in statistics and become inactive in this group.

Such co-evolutionary nature of user-item interactions raises very important questions on how to learn them from the increasingly available data. However, existing methods either treat the temporal user-item interactions data as a static graph or use epoch based methods such as tensor factorization to learn the latent features (Chi & Kolda, 2012; Koren, 2009; Yang et al., 2011). These methods are not able to capture the fine grained temporal dynamics of user-item interactions. Recent point process based models treat time as a random variable and improves over the traditional methods significantly (Du et al., 2015; Wang et al., 2016b). However, these works make strong assumptions

---

*Authors have equal contributions.

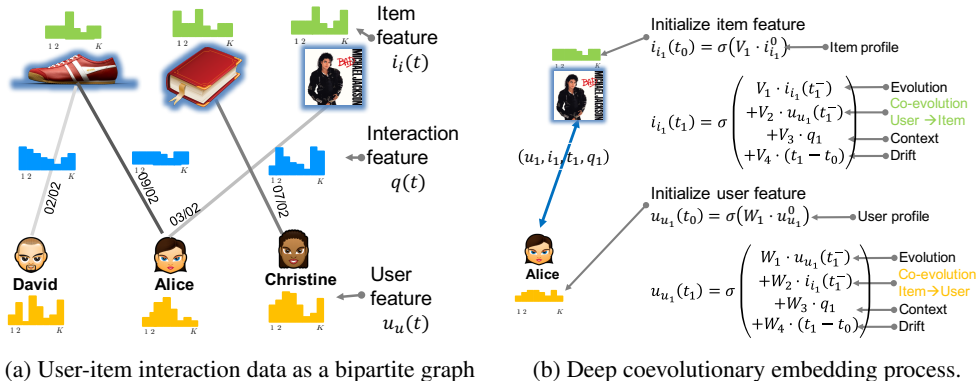

(a) User-item interaction data as a bipartite graph  (b) Deep coevolutionary embedding process.

Figure 1: Model illustration. (a) User-item interaction events data. Each edge stands for a tuple and contains the information of user, item, interaction time, and interaction feature. (b) The latent feature of the user and item are updated at each event time, by a nonlinear activation function $\sigma(\cdot)$ and contain four terms: self evolution, co-evolution, context (interaction feature), and self drift.

about the function form of the generative processes, which may not reflect the reality or accurate enough to capture the complex and *nonlinear* user-item influence in real world.

In this paper, we propose a recurrent coevolutionary feature embedding process framework. It combines recurrent neural network (RNN) with point process models, and efficiently captures the co-evolution of user-item features. Our model can automatically find an efficient representation of the underlying user and item latent feature without assuming a fixed parametric forms in advance. Figure 1 summarizes our framework. In particular, our work makes the following contributions:

- **Novel model.** We propose a novel model that captures the *nonlinear* co-evolution nature of users' and items' embeddings. It assigns an evolving feature embedding process for each user and item, and the co-evolution of these latent feature processes is modeled with two parallel components: (i) *item → user* component, a user's latent feature is determined by the nonlinear embedding of latent features of the items he interacted with; and (ii) *user → item* component, an item's latent features are also determined by the latent features of the users who interact with the item.
- **Technical Challenges.** We use RNN to parametrize the interdependent and intertwined user and item embeddings. The increased flexibility and generality further introduces technical challenges on how to train RNN on the co-evolving graphs. The co-evolution nature of the model makes the samples inter-dependent and not identically distributed, which is contrary to the assumptions in the traditional setting and significantly more challenging. We are the first to propose an efficient stochastic training algorithm that makes the BTPP tractable in the co-evolving graph.
- **Strong performance.** We evaluate our method over multiple datasets, verifying that our method can lead to significant improvements in user behavior prediction compared to previous state-of-the-arts. Precise time prediction is especially novel and not possible by most prior work.

## 2 RELATED WORK

Recent work predominantly fix the latent features assigned to each user and item (Salakhutdinov & Mnih, 2008; Chen et al., 2009; Agarwal & Chen, 2009; Ekstrand et al., 2011; Koren & Sill, 2011; Yang et al., 2011; Yi et al., 2014; Wang & Pal, 2015). In more sophisticated methods, the time is divided into epochs, and static latent feature models are applied to each epoch to capture some temporal aspects of the data (Koren, 2009; Karatzoglou et al., 2010; Xiong et al., 2010; Karatzoglou et al., 2010; Xiong et al., 2010; Chi & Kolda, 2012; Gultekin & Paisley, 2014; Charlin et al., 2015; Preeti Bhargava, 2015; Gopalan et al., 2015; Hidasi & Tikk, 2015; Wang et al., 2016a). For such methods, it is not clear how to choose the epoch length parameter. First, different users may have very different timescale when they interact with those service items, making it difficult to choose a unified epoch length. Second, it is not easy for these methods to answer time-sensitive queries such as when a user will return to the service item. The predictions are only in the resolution of the chosen epoch length. Recently, (Du et al., 2015) proposed a low-rank point process based model for time-sensitive recommendations from recurrent user activities. However, it fails to capture the heterogeneous coevolutionary properties of user-item interactions. Wang et al. (2016b) models the co-evolutionary property, but uses a simple linear representation of the users' and items' latent features, which might not be expressive enough to capture the real world patterns. As demonstrated in Du et al. (2016),

the nonlinear RNN is quite flexible to approximate many point process models. Also we will show that, our model only has O(#user + #item) regardless of RNN related parameters, and can also be potentially applied to online setting.

In the deep learning community, (Wang et al., 2015a) proposed a hierarchical Bayesian model that jointly performs learning for the content features and collaborative filtering for the ratings matrix. (Hidasi et al., 2016) applied RNN and adopt item-to-item recommendation approach with session based data. (Tan et al., 2016) improved this model with techniques like data augmentation, temporal change adaptation. (Ko et al., 2016) proposed collaborative RNN that extends collaborative filtering method to capture history of user behavior. Specifically, they used static global latent factors for items and assign separate latent factors for users that are dependent on their past history. (Song et al., 2016) extended the deep semantic structured model to capture multi-granularity temporal preference of users. They use separate RNN for each temporal granularity and combine them with feed forward network which models users' and items' long term static features. However, none of these works model the coevolution of users' and items' latent features and are still extensions of epoch based methods. Our work is unique since we explicitly treat time as a random variable and captures the coevolution of users' and items' latent features using temporal point processes. Finally, our work is inspired from the recurrent marked temporal point process model (Du et al., 2016). However, this work only focuses on learning a one-dimension point process. Our work is significantly different since we focus on the recommendation system setting with the novel idea of feature coevolution and we use multi-dimensional point processes to capture user-item interactions.

## 3 BACKGROUND ON TEMPORAL POINT PROCESSES

A temporal point process (Cox & Isham, 1980; Cox & Lewis, 2006; Aalen et al., 2008) is a random process whose realization consists of a list of discrete events localized in time, $\{t_i\}$ with $t_i \in \mathbb{R}^+$. Equivalently, a given temporal point process can be represented as a counting process, $N(t)$, which records the number of events before time $t$. An important way to characterize temporal point processes is via the conditional intensity function $\lambda(t)$, a stochastic model for the time of the next event given all the previous events. Formally, $\lambda(t)\mathrm{d}t$ is the conditional probability of observing an event in a small window $[t, t+\mathrm{d}t)$ given the history $\mathcal{H}(t)$ up to $t$ and that the event has not happen before $t$, *i.e.*,

$$\lambda(t)\mathrm{d}t := \mathbb{P}\left\{\text{event in } [t, t+\mathrm{d}t)|\mathcal{H}(t)\right\} = \mathbb{E}[\mathrm{d}N(t)|\mathcal{H}(t)],$$

where one typically assumes that only one event can happen in a small window of size $\mathrm{d}t$, *i.e.*, $\mathrm{d}N(t) \in \{0, 1\}$. Then, given a time $t \geqslant 0$, we can also characterize the conditional probability that no event happens during $[0, t)$ as: $S(t) = \exp\left(-\int_0^t \lambda(\tau)\,\mathrm{d}\tau\right)$ and the conditional density that an event occurs at time $t$ is defined as

$$f(t) = \lambda(t)\,S(t) \tag{1}$$

The function form of the intensity $\lambda(t)$ is often designed to capture the phenomena of interests. Some commonly used form includes:

- Hawkes processes (Hawkes, 1971; Wang et al., 2016c), whose intensity models the mutual excitation between events, *i.e.*, $\lambda(t) = \mu + \alpha \sum_{t_i \in \mathcal{H}(t)} \kappa_\omega(t - t_i)$, where $\kappa_\omega(t) := \exp(-\omega t)$ is an exponential triggering kernel, $\mu \geqslant 0$ is a baseline intensity. Here, the occurrence of each historical event increases the intensity by a certain amount determined by the kernel $\kappa_\omega$ and the weight $\alpha \geqslant 0$, making the intensity history dependent and a stochastic process by itself.
- Rayleigh process, whose intensity function is $\lambda(t) = \alpha t$, where $\alpha > 0$ is the weight parameter.

## 4 RECURRENT COEVOLUTIONARY FEATURE EMBEDDING PROCESSES

In this section, we present the generative framework for modeling the temporal dynamics of user-item interactions. We first use RNN to explicitly capture the co-evolving nature of users' and items' latent feature. Then, based on the compatibility between the users' and items' latent feature, we model the user-item interactions by a multi-dimensional temporal point process. We further parametrize the intensity function by the compatibility between users' and items' latent features.

### 4.1 EVENT REPRESENTATION

Given $m$ users and $n$ items, we denote the ordered list of $N$ observed events as $\mathcal{O} = \{e_j = (u_j, i_j, t_j, \boldsymbol{q}_j)\}_{j=1}^N$ on time window $[0, T]$, where $u_j \in \{1, \ldots, m\}$, $i_j \in \{1, \ldots, n\}$, $t_j \in \mathbb{R}^+$, $0 \leqslant t_1 \leqslant t_2 \ldots \leqslant T$. This represents the interaction between user $u_j$, item $i_j$ at time $t_j$, with the interaction context $\boldsymbol{q}_j \in \mathbb{R}^d$. Here $\boldsymbol{q}_j$ can be a high dimension vector such as the text review, or

simply the embedding of static user/item features such as user's profile and item's categorical features. For notation simplicity, we define $\mathcal{O}^u = \{e_j^u = (i_j^u, t_j^u, \boldsymbol{q}_j^u)\}$ as the ordered listed of all events related to user $u$, and $\mathcal{O}^i = \{e_j^i = (u_j^i, t_j^i, \boldsymbol{q}_j^i)\}$ as the ordered list of all events related to item $i$. We also set $t_0^i = t_0^u = 0$ for all the users and items. $t_k-$ denotes the time point just before time $t_k$.

## 4.2 RECURRENT FEATURE EMBEDDING PROCESSES

We associate feature embeddings $\boldsymbol{u}_u(t) \in \mathbb{R}^k$ with each user $u$ and $\boldsymbol{i}_i(t) \in \mathbb{R}^k$ with each item $i$. These features represent the subtle properties which cannot be directly observed, such as the interests of a user and the semantic topics of an item. Specifically, we model the *drift, evolution, and co-evolution* of $\boldsymbol{u}_u(t)$ and $\boldsymbol{i}_i(t)$ as a piecewise constant function of time that has jumps only at event times. Specifically, we define:

**User latent feature embedding process.** For each user $u$, the corresponding embedding after user $u$'s $k$-th event $e_k^u = (i_k^u, t_k^u, \boldsymbol{q}_k^u)$ can be formulated as:

$$\boldsymbol{u}_u(t_k^u) = \sigma\bigg( \underbrace{\boldsymbol{W}_1(t_k^u - t_{k-1}^u)}_{\text{temporal drift}} + \underbrace{\boldsymbol{W}_2\boldsymbol{u}_u(t_{k-1}^u)}_{\text{self evolution}} + \underbrace{\boldsymbol{W}_3\boldsymbol{i}_{i_k}(t_k^u-)}_{\text{co-evolution: item feature}} + \underbrace{\boldsymbol{W}_4\boldsymbol{q}_k^{u,i_k}}_{\text{interaction feature}} \bigg) \quad (2)$$

**Item latent feature embedding process.** For each item $i$, we specify $\boldsymbol{i}_i(t)$ at time $t_k^i$ as:

$$\boldsymbol{i}_i(t_k^i) = \sigma\bigg( \underbrace{\boldsymbol{V}_1(t_k^i - t_{k-1}^i)}_{\text{temporal drift}} + \underbrace{\boldsymbol{V}_2\boldsymbol{i}_i(t_{k-1}^i)}_{\text{self evolution}} + \underbrace{\boldsymbol{V}_3\boldsymbol{u}_{u_k}(t_k^i-)}_{\text{co-evolution: item feature}} + \underbrace{\boldsymbol{V}_4\boldsymbol{q}_k^{i,u_k}}_{\text{interaction feature}} \bigg) \quad (3)$$

where $t-$ means the time point just before time $t$, $\boldsymbol{W}_4, \boldsymbol{V}_4 \in \boldsymbol{R}^{k \times d}$ are the embedding matrices mapping from the explicit high-dimensional feature space into the low-rank latent feature space and $\boldsymbol{W}_1, \boldsymbol{V}_1 \in \mathbb{R}^k$, $\boldsymbol{W}_2, \boldsymbol{V}_2, \boldsymbol{W}_3, \boldsymbol{V}_3 \in \mathbb{R}^{k \times k}$ are weights parameters. $\sigma(\cdot)$ is the nonlinear activation function, such as commonly used Tanh or Sigmoid for RNN. For simplicity, we use basic recurrent neural network to formulate the recurrence, but it is also straightforward to extend it using GRU or LSTM to gain more expressive power. Figure 1 summarizes the basic setting of our model.

Here both the user and item's feature embedding processes are piecewise constant functions of time and *only updated if an interaction event happens*. A user's attribute changes only when he has a new interaction with some item. For example, a user's taste for music changes only when he listens to some new or old musics. Also, an item's attribute changes only when some user interacts with it. Different from Chen et al. (2013) who also models the time change with piecewise constant function, but their work has no coevolve modeling, and is not capable of predicting the future time point.

Next we discuss the rationale of each term in detail:

- **Temporal drift**. The first term is defined based on the time difference between consecutive events of specific user or item. It allows the basic features of users (*e.g.*, a user's self-crafted interests) and items (*e.g.*, textual categories and descriptions) to smoothly drift through time. Such changes of basic features normally are caused by external influences.
- **Self evolution**. The current user feature should also be influenced by its feature at the earlier time. This captures the intrinsic evolution of user/item features. For example, a user's current taste should be more or less similar to his/her tastes two days ago.
- **User-item coevolution**. Users' and items' latent features can mutually influence each other. This term captures the two parallel processes. First, a user's embedding is determined by the latent features of the items he interacted with. At each time $t_k$, the latent item feature is $\boldsymbol{i}_{i_k}(t_k^u-)$. We capture both the temporal influence and feature of each history item as a latent embedding. Conversely, an item's embedding is determined by the feature embedding of the user who just interacts with the item.
- **Evolution with interaction features**. Users' and items' features can evolve and be influenced by the characteristics of their interactions. For instance, the genre changes of movies indicate the changing tastes of users. The theme of a chatting-group can be easily shifted to certain topics of the involved discussions. In consequence, this term captures the influence of the current interaction features to the changes of the latent user (item) features.
- **Interaction feature**. This is the additional information happened in the user-item interactions. For example, in online discussion forums such as Reddit, the interaction features are the posts and comments. In online review sites such as Yelp, it is the reviews of the businesses.

To summarize, each feature embedding process evolves according to the respective base temporal user (item) features and also are mutually dependent on each other due to the endogenous influences from the interaction features and the entangled latent features.

### 4.3 USER-ITEM INTERACTIONS AS TEMPORAL POINT PROCESSES

For each user, we model the recurrent occurrences of all users interaction with all items as a multi-dimensional temporal point process, with each user-item pair as one dimension. In particular, the intensity function in the $(u, i)$-th dimension (user $u$ and item $i$) is modeled as a Rayleigh process:

$$\lambda^{u,i}(t|t') = \underbrace{\exp\left(\boldsymbol{u}_u(t')^\top \boldsymbol{i}_i(t')\right)}_{\text{user-item compatibility}} \cdot \underbrace{(t - t')}_{\text{time lapse}} \tag{4}$$

where $t > t'$, and $t'$ is the last time point where either user $u$'s embedding or item $i$'s embedding changes before time $t$. The rationale behind this formulation is three-fold:

- *Time as a random variable.* Instead of discretizing the time into epochs as traditional methods (Charlin et al., 2015; Preeti Bhargava, 2015; Gopalan et al., 2015; Hidasi & Tikk, 2015; Wang et al., 2016a), we explicitly model the timing of each interaction event as a random variable, which naturally captures the heterogeneity of the temporal interactions between users and items.
- *Short term preference.* The probability for user $u$ to interact with item $i$ depends on the compatibility of their instantaneous embeddings, which is evaluated through the inner product at the last event time $t'$. Because $\boldsymbol{u}_u(t)$ and $\boldsymbol{i}_i(t)$ co-evolve through time, their inner-product measures a general representation of the cumulative influence from the past interactions to the occurrence of the current event. The $\exp(\cdot)$ function ensures the intensity is positive and well defined.
- *Rayleigh time distribution.* The user and item embeddings are *piecewise constant*, and we use the time lapse term to make the intensity *piecewise linear*. This form leads to a Rayleigh distribution for the time intervals between consecutive events in each dimension. It is well-adapted to modeling fads, where the event-happening likelihood $f(\cdot)$ in (1) rises to a peak and then drops extremely rapidly. Furthermore, it is computationally easy to obtain an analytic form of $f(\cdot)$. One can then use $f(\cdot)$ to make item recommendation by finding the dimension that $f(\cdot)$ reaches the peak.

With the parameterized intensity function, we can further estimate the parameters using maximum likelihood estimation of all events. The joint negative log-likelihood is (Daley & Vere-Jones, 2007):

$$\ell = -\underbrace{\sum_{j=1}^{N} \log\left(\lambda^{u_j, i_j}(t_j|t'_j)\right)}_{\text{intensity of interaction event}} + \underbrace{\sum_{u=1}^{m}\sum_{i=1}^{n}\int_{0}^{T} \lambda^{u,i}(\tau|\tau')\, d\tau}_{\text{survival probability of event not happened}} \tag{5}$$

The rationale of the objective two-fold: (i) the negative intensity summation term ensures the probability of all interaction events is maximized; (ii) the second survival probability term penalizes the *non-presence* of an interaction between all possible user-item pairs on the observation window. Hence, our framework not only explains why an event happens, but also why an event did not happen.

### 5 PARAMETER LEARNING

In this section, we propose an efficient algorithm to learn the parameters $\{\boldsymbol{V}_i\}_{i=1}^{4}$ and $\{\boldsymbol{W}_i\}_{i=1}^{4}$. The batch objective function is presented in (5). The Back Propagation Through Time (BPTT) is the standard way to train a RNN. To make the back propagation tractable, one typically needs to do truncation during training. However, due to the novel co-evolutionary nature of our model, all the events are related to each other by the user-item bipartite graph (Figure 2), which makes it hard to decompose.

Hence, in sharp contrast to works (Hidasi et al., 2016; Du et al., 2016) in sequential data where one can easily break the sequences into multiple segments to make the BPTT trackable, it is a *challenging* task to design BPTT in our case. To efficiently solve this problem, we first order all the events globally and then do mini-batch training in a sliding window fashion. Each time when conducting feed forward and back propagation, we take the consecutive events within current sliding window to build the computational graph. Thus in our case the truncation is on the global timeline, instead over individual independent sequences as in prior works.

Next, we explain our procedure in detail. Given a mini-batch of $M$ ordered events $\tilde{\mathcal{O}} = \{e_j\}_{j=1}^{M}$, we set the time span to be $[T_0 = t_1, T = t_M]$. Below we show how to compute the intensity and survival probability term in the objective function (5) respectively.

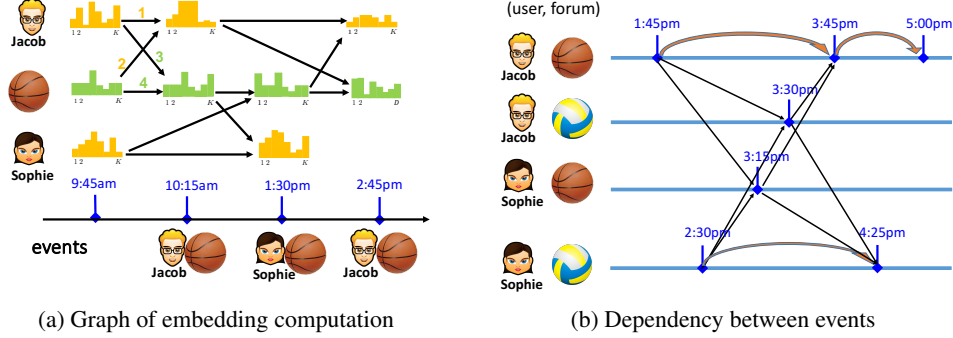

(a) Graph of embedding computation  (b) Dependency between events

Figure 2: Intensity computation. (a) Each arrow means the flow of feature embedding computation, *e.g.*, Jacob interacts with basketball at 10:15am. Then the embeddings are updated: his feature at 10:15 am is influenced by his feature and the basketball feature at 9:45am (arrow 1 and 2); the basketball's feature is influenced by Jacob's feature and its feature (arrow 3 and 4). (b) The events dependency for two users and two forums (items). It shows how event at one dimension influence other dimensions. Each orange arrow represents the dependency within each dimension, and the black arrow denotes the cross-dimension dependency, *e.g.*, Sophie interacts with volleyball at 2:30pm, and this event changes the volleyball embedding, thus will affect Jacob's visit at 3:30pm.

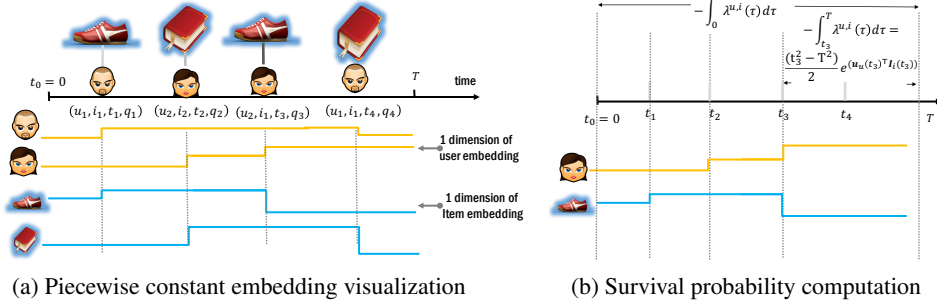

(a) Piecewise constant embedding visualization  (b) Survival probability computation

Figure 3: Survival probability computation. (a) A user or item's feature embedding is piecewise constant and will change *only* after an interaction event happens. Only one dimension of the feature embedding is shown. (b) Survival probability for a user-item pair $(u, i)$. The integral $\int_0^T \lambda^{u,i}(\tau|\tau')\mathrm{d}\tau$ is decomposed into 4 inter-event intervals separated by $\{t_0, \cdots, t_3\}$, with close form on each interval.

**Computing the intensity function.** Each time when a new event $e_j$ happens between $u_j$ and $i_j$, their corresponding feature embeddings will evolve according to a computational graph, as illustrated in Figure 2a. Due to the change of feature embedding, all the dimensions related to $u_j$ or $i_j$ will be influenced and the intensity function for that dimension will change consequently. Such cross-dimension influence dependency is shown in Figure 2b. In our implementation, we first compute the corresponding intensity $\lambda^{u_j, i_j}(t_j|t_j')$ according to (4), and then update the embedding of $u_j$ and $i_j$. This operation takes $O(M)$ complexity, and is independent to the number of users or items.

**Computing the survival function.** To compute the survival probability $-\int_{T_0}^T \lambda^{u,i}(\tau|\tau')\mathrm{d}\tau$ for each pair $(u, i)$, we first collect all the time stamps $\{t_k\}$ that have events related to either $u$ or $i$. For notation simplicity, let $|\{t_k\}| = n_{u,i}$ and $t_1 = T_0, t_{n_{u,i}} = T$. Since the embeddings are piecewise constant, the corresponding intensity function is piecewise linear, according to (4). Thus, the integration is decomposed into each time interval where the intensity is constant, *i.e.*,

$$\int_{T0}^T \lambda^{u,i}(\tau|\tau')\mathrm{d}\tau = \sum_{k=1}^{n_{u,i}-1} \int_{t_k}^{t_{k+1}} \lambda^{u,i}(\tau|\tau')\mathrm{d}\tau = \sum_{k=1}^{n_{u,i}-1} (t_{k+1}^2 - t_k^2) \exp\left(\boldsymbol{u}_u(t_k)^\top \boldsymbol{i}_i(t_k)\right) \quad (6)$$

Figure 3 visualizes the computation. Although the survival probability term exists in close form, we still need to solve two challenges. First, it is still expensive to compute it for each user item pair. Moreover, since the user-item interaction bipartite graph is very sparse, it is not necessary to monitor each dimension in the stochastic training setting. To speed up the computation, we propose a novel random-sampling scheme as follows.

Note that the intensity term in the objective function (5) tries to maximize the inner product between user and item that has interaction event, while the survival term penalize over all other pairs of inner

Table 1: Comparison with different methods.

| Method | DeepCoevolve | LowRankHawkes | Coevolving | PoissonTensor | TimeSVD++ | FIP | STIC |
|---|---|---|---|---|---|---|---|
| Continuous time | √ | √ | √ | | | | √ |
| Predict Item | √ | √ | √ | √ | √ | √ | |
| Predict Time | √ | √ | √ | √ | | | √ |
| Computation | RNN | Factorization | Factorization | Factorization | Factorization | Factorization | HMM |

products. We observe that this is similar to Softmax computing for classification problem. Hence, inspired by the noise-contrastive estimation method (Gutmann & Hyvärinen, 2012) that is widely used in language models (Mnih & Kavukcuoglu, 2013), we keep the dimensions that have events on them, while randomly sample dimensions without events in current mini-batch.

The second challenge lies in the fact that the user-item interactions vary a lot across mini-batches, hence the corresponding computational graph also changes greatly. To make the learning efficient, we use the graph embedding framework (Dai et al., 2016) which allows training deep learning models where each term in the objective has a different computational graphs but with shared parameters. The Adam Optimizer (Kingma & Ba, 2014) together with gradient clip is used in our experiment.

## 6 EXPERIMENTS

We evaluate our model on real-world datasets. For each sequence of user activities, we use all the events up to time $T \cdot p$ as the training data, and the rest events as the testing data, where $T$ is the observation window. We tune the latent rank of other baselines using 5-fold cross validation with grid search. We vary the proportion $p \in \{0.7, 0.72, 0.74, 0.76, 0.78\}$ and report the averaged results over five runs on two tasks (we will release code and data once published):

- *Item prediction*. At each test time $t$, we predict the item that the user $u$ will interact with. We rank all the items in the descending order of the conditional density $f^{u,i}(t) = \lambda^{u,i}(t)S^{u,i}(t)$. We report the Mean Average Rank (MAR) of each test item at the test time. Ideally, the item associated with the test time $t$ should rank one, hence smaller value indicates better predictive performance.
- *Time prediction*. We predict the expected time when a testing event will occur between a given user-item pair. Using Rayleigh distribution, it is given by $\mathbb{E}_{t \sim f^{u,i}(t)}(t) = \sqrt{\frac{\pi}{2 \exp(\boldsymbol{u}_u(t-)^\top \boldsymbol{i}_i(t-))}}$. We report the Mean Absolute Error (MAE) between the predicted and true time.

### 6.1 COMPETITORS

We compared our DEEPCOEVOLVE with the following methods. Table 1 summarizes the differences.

- **LowRankHawkes** (Du et al., 2015): This is a low rank Hawkes process model which assumes user-item interactions to be independent of each other and does not capture the co-evolution of user and item features.
- **Coevolving** (Wang et al., 2016b): This is a multi-dimensional point process model which uses a simple linear embedding to model the co-evolution of user and item features.
- **PoissonTensor** (Chi & Kolda, 2012): Poisson Tensor Factorization has been shown to perform better than factorization methods based on squared loss (Karatzoglou et al., 2010; Xiong et al., 2010; Wang et al., 2015b) on recommendation tasks. The performance for this baseline is reported using the average of the parameters fitted over all time intervals.
- **TimeSVD++** (Koren, 2009) and **FIP** (Yang et al., 2011): These two methods are only designed for explicit ratings, the implicit user feedbacks (in the form of a series of interaction events) are converted into the explicit ratings by the respective frequency of interactions with users.
- **STIC** (Kapoor et al., 2015): it fits a semi-hidden markov model (HMM) to each observed user-item pair and is only designed for time prediction.

### 6.2 DATASETS

We use three real world datasets as follows.

- **IPTV.** It contains 7,100 users' watching history of 385 TV programs in 11 months (Jan 1 - Nov 30 2012), with around 2M events, and 1,420 movie features (including 1,073 actors, 312 directors, 22 genres, 8 countries and 5 years).
- **Yelp.** This data was available in Yelp Dataset challenge Round 7. It contains reviews for various businesses from October, 2004 to December, 2015. The dataset we used here contains 1,005 users and 47,924 businesses, with totally 291,716 reviews.

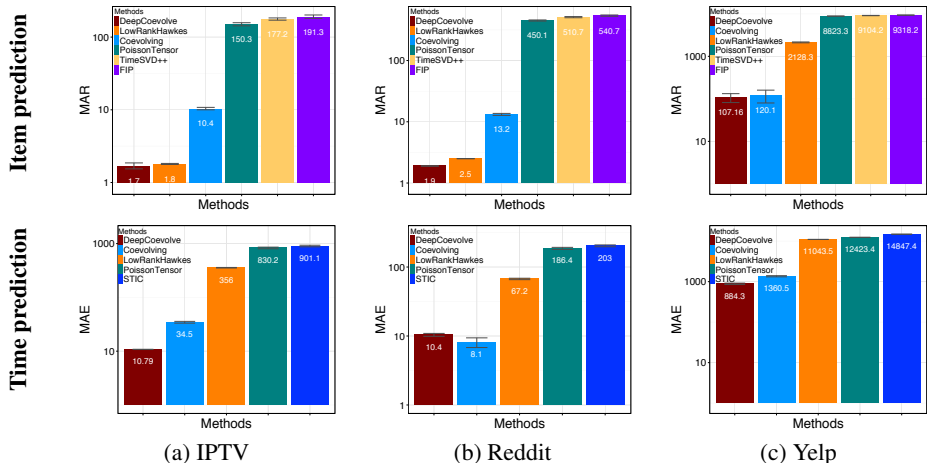

Figure 4: Prediction results on three real world datasets.

- **Reddit.** We collected discussion related data on different subreddits (groups) for the month of January 2014. We filtered all bot users' and their posts from this dataset. Furthermore, we randomly selected 1,000 users, 1,403 groups, and 10,000 discussion events.

## 6.3 PREDICTION RESULTS

Figure 4 shows that DEEPCOEVOLVE significantly outperforms both epoch-based baselines and state-of-arts point process based methods. LOWRANKHAWKES has good performance on item prediction but not on time prediction, while COEVOLVING has good performance on time prediction but not on item prediction. We discuss the performance regarding the two metrics below.

**Item prediction.** Note that the best possible MAR one can achieve is 1, and our method gets quite accurate results: with the value of 1.7 on IPTV and 1.9 on Reddit. Note LOWRANKHAWKES achieves comparable item prediction performance, but not as good on the time prediction task. We think the reason is as follows. Since one only need the rank of conditional density $f(\cdot)$ in (1) to conduct item prediction, LOWRANKHAWKES may still be good at differentiating the conditional density function, but could not learn its actual value accurately, as shown in the time prediction task where the value of the conditional density function is needed for precise prediction.

**Time prediction.** The second row of Figure 4 shows that DEEPCOEVOLVE outperforms other methods. Compared with LOWRANKHAWKES that achieves comparable time predication performance, $6\times$ improvement on Reddit, it has $10\times$ improvement on Yelp, and $30\times$ improvement on IPTV. The time unit is hour. Hence it has 2 weeks accuracy improvement on IPTV and 2 days on Reddit. This is important for online merchants to make time sensitive recommendations. An intuitive explanation is that our method accurately captures the *nonlinear* pattern between user and item interactions. The competitor LOWRANKHAWKES assumes specific parametric forms of the user-item interaction process, hence may not be accurate or expressive enough to capture real world temporal patterns. Furthermore, it models each user-item interaction dimension independently, which may lose the important affection from user's interaction with other items while predicting the current item's reoccurrence time. Our work also outperforms COEVOLVING, *e.g.*, with around $3\times$ MAE improve on IPTV. Moreover, the item prediction performance is also much better than COEVOLVING. It shows the importance of using RNN to capture the nonlinear embedding of user and item latent features, instead of the simple parametrized linear embedding in COEVOLVING.

## 6.4 INSIGHT OF RESULTS

We will look deeper and provide rationale behind the prediction results in the following two subsections. First, to understand the difficulty of conducting prediction tasks in each dataset, we study their different sparsity properties. For the multidimensional point process models, the fewer events we observe in each dimension, the more sparse the dataset is. Our approach alleviates the sparsity problem via the modeling of dependencies among dimensions, thus is consistently doing better than other baseline algorithms.

Next, we fix one dataset and evaluate how different levels of sparsity in training data influences each algorithm's performance.

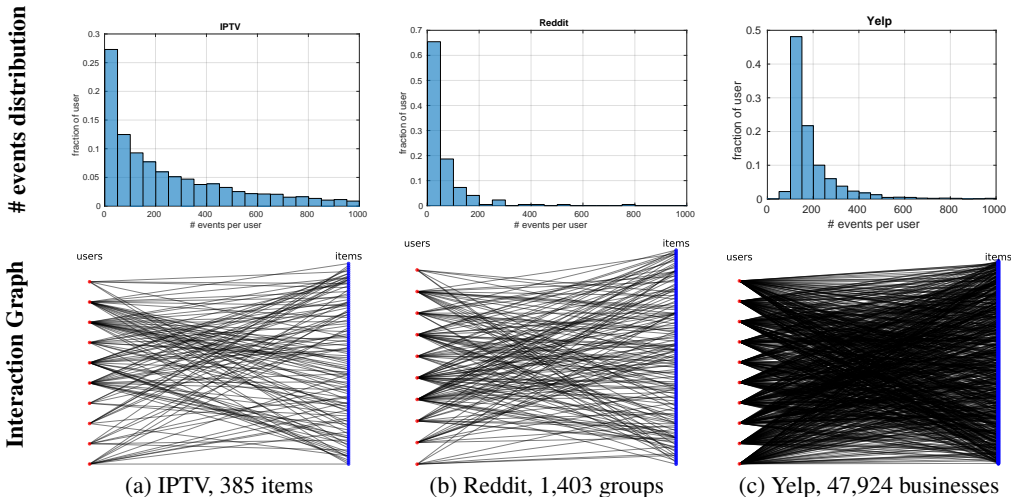

Figure 5: Visualization of the sparsity property in each dataset. The first row shows the distribution of number of events per user. The second row shows the user-item interaction graph. It is generated as follows. For each dataset, we randomly pick 10 users with 100 history events each user and collect all items they have interacted with. The interaction graph itself is a bipartite graph, and we put users on left side, and items on the right side.

### 6.4.1 UNDERSTANDING THE DATASETS

We visualize the three datasets in Figure 5 according to (i) the number of events per user, and (ii) the user-item interaction graph.

**Sparsity in terms of the number of events per user.** Typically, the more user history data we have, the better results we will obtain in the prediction tasks. We can see in IPTV dataset, users typically have longer length of history than the users in Reddit and Yelp datasets. Thus our algorithm and all other baseline methods have their best performance on this dataset. However, the Reddit dataset and Yelp dataset are hard to tell the performance based only on the distribution of history length, thus we do a more detailed visualization.

**Sparsity in terms of diversity of items to recommend.** From the bipartite graph, it is easy to see that Yelp dataset has higher density than the other two datasets. The density of the interaction graph reflects the variety of history per each user. For example, the users in IPTV only has 385 programs to watch, but they can have 47,924 businesses to choose in Yelp dataset. Also, the Yelp dataset has 9 times more items than IPTV and Reddit dataset in the bipartite graph. This means the users in Yelp dataset has more diverse tastes than users in other two datasets. This is because if users has similar tastes, the distinct number of items in the union of their history should be small.

Based on the above two facts, we can see Yelp dataset is the most sparse, since it has shorter length of history per user, and much more diversity of the items, it is not surprising that this dataset is much harder than the other IPTV and Reddit dataset.

### 6.4.2 ROBUSTNESS OF THE ALGORITHM

With the case study on the most challenging Yelp dataset, we further evaluate how each algorithm performs with lower level of sparsity as compared to the one used in Figure 4 (c).We use this to demonstrate that our work is most robust and performs well across different levels of sparsity.

We first create Yelp100, a more dense dataset, by filtering the original Yelp dataset to keep the top 100 users. Each user would have at least 200 events. Figure 6 (a) shows the statistics of this dataset. On average the users have more history events than the original Yelp dataset in Figure 5(c).

On this dense dataset, Figure 6 (b) and (c) show that all the algorithms' performances improve with more history events, comparing to the performance in original Yelp dataset. For example, LOWRANKHAWKES has similar rank prediction results as our DEEPCOEVOLVE on this dense dataset. However, as the dataset becomes sparse, the performance of LOWRANKHAWKES drops significantly, as shown in Figure 4(c). For example, the rank prediction error goes from 90 to 2128, and the

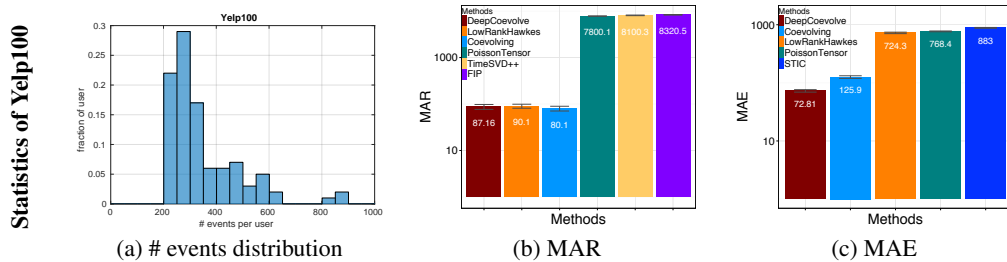

(a) # events distribution  (b) MAR  (c) MAE

Figure 6: Comparison of performance with different amount of history.

time error goes from 724 to 11043.5. We think it is because this model relies more on the history information per each user-item pair.

On the contrary, our DEEPCOEVOLVE still has superior performance with such high level of sparsity. The rank error only changes from 87 to 107, and the time error changes from 72 to 884 as the data becomes sparse. It shows that our work is the *most robust to the sparsity in the data*. We think it is because our work accurately captures the nonlinear multidimensional dependencies between users and items latent features.

## 7  CONCLUSION

We have proposed an efficient framework to model the *nonlinear* co-evolution nature of users' and items' latent features. Moreover, the user and item's evolving and co-evolving processes are captured by the RNN. It is based on temporal point processes and models time as a random variable. Hence it is in sharp contrast to prior epoch based works. We demonstrate the superior performance of our method on both the time and item prediction task, which is not possible by most prior work. Future work includes extending to other social applications, such as group dynamics in message services.

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

# A  DETAILS ON GRADIENT COMPUTATION

**Computing gradient.** For illustration purpose, we here use Sigmoid as the nonlinear activation function $\sigma$. In order to get gradient with respect to parameter $\boldsymbol{W}$'s, we first compute gradients with respect to each varying points of embeddings. For user $u$'s embedding after his $k$-th event, the corresponding partial derivatives are computed by:

$$
\frac{\partial \ell}{\partial \boldsymbol{u}_u(t_k^u)} = \underbrace{-\boldsymbol{i}_{i_k^u}}_{\text{from intensity}} + \underbrace{\sum_{i=1}^{n} \frac{\partial \int_{t_k^u}^{t_{k+1}^u} \lambda^{u,i}(\tau|\tau')d\tau}{\partial \boldsymbol{u}_u(t_k^u)}}_{\text{from survival}} + \underbrace{\frac{\partial \ell}{\partial \boldsymbol{u}_u(t_{k+1}^u)} \odot (1 - \boldsymbol{u}_u(t_{k+1}^u)) \odot \boldsymbol{u}_u(t_{k+1}^u)\boldsymbol{W}_2}_{\text{from user } u\text{'s next embedding}}
$$

$$
+ \underbrace{\frac{\partial \ell}{\partial \boldsymbol{i}_{i_{k+1}^u}(t_{k+1}^u)} \odot (1 - \boldsymbol{i}_{i_{k+1}^u}(t_{k+1}^u)) \odot \boldsymbol{i}_{i_{k+1}^u}(t_{k+1}^u)}_{\text{from user } u\text{'s next item embedding}}
$$

where $\odot$ denotes element-wise multiplication.

The gradient coming from the second term (*i.e.*, the survival term) is also easy to compute, since the Rayleigh distribution has closed form of survival function. For a certain item $i$, if its feature doesn't changed between time interval $[t_k^u, t_{k+1}^u]$, then we have

$$
\frac{\partial \int_{t_k^u}^{t_{k+1}^u} \lambda^{u,i}(\tau|\tau')d\tau}{\partial \boldsymbol{u}_u(t_k^u)} = \frac{(t_{k+1}^u - t_k^u)^2}{2} \exp\left(\boldsymbol{u}_u(t_k^u)^\top \boldsymbol{i}_i(t_k^u)\boldsymbol{i}_i(t_k^u)\right) \tag{7}
$$

On the other hand, if the embedding of item $i$ changes during this time interval, then we should break this interval into segments and compute the summation of gradients in each segment in a way similar to (7). Thus, we are able to compute the gradients with respect to $\boldsymbol{W}_i, i \in \{1, 2, 3, 4\}$ as follows.

$$
\frac{\partial \ell}{\partial \boldsymbol{W}_1} = \sum_{u=1}^{m} \sum_{k} \frac{\partial \ell}{\partial \boldsymbol{u}_u(t_k^u)} \odot (\boldsymbol{i} - \boldsymbol{u}_u(t_k^u)) \odot \boldsymbol{u}_u(t_k^u)(t_k^u - t_{k-1}^u)
$$

$$
\frac{\partial \ell}{\partial \boldsymbol{W}_2} = \sum_{u=1}^{m} \sum_{k} \left(\frac{\partial \ell}{\partial \boldsymbol{u}_u(t_k^u)} \odot (\boldsymbol{i} - \boldsymbol{u}_u(t_k^u)) \odot \boldsymbol{u}_u(t_k^u)\right) \boldsymbol{u}_u(t_{k-1}^u)^\top
$$

$$
\frac{\partial \ell}{\partial \boldsymbol{W}_3} = \sum_{u=1}^{m} \sum_{k} \left(\frac{\partial \ell}{\partial \boldsymbol{u}_u(t_k^u)} \odot (\boldsymbol{i} - \boldsymbol{u}_u(t_k^u)) \odot \boldsymbol{u}_u(t_k^u)\right) \boldsymbol{i}_{i_k}(t_k^u-)^\top
$$

$$
\frac{\partial \ell}{\partial \boldsymbol{W}_4} = \sum_{u=1}^{m} \sum_{k} \left(\frac{\partial \ell}{\partial \boldsymbol{u}_u(t_k^u)} \odot (\boldsymbol{i} - \boldsymbol{u}_u(t_k^u)) \odot \boldsymbol{u}_u(t_k^u)\right) \boldsymbol{q}_k^{u,i_k}
$$

Since the items are treated symmetrically as users, the corresponding derivatives can be obtained in a similar way.

