# Peer review of "Recurrent Coevolutionary Feature Embedding Processes for Recommendation"

_ICLR 2017 — rejected_

[Public Comment · (anonymous) · 09 Dec 2016]
**Already published?**

Nice work, but is seems to me that it was already published at DLRS 2016 in September (

[Official Review · AnonReviewer1 · rating 6 · confidence 3 · 15 Dec 2016]

The paper seeks to predict user events (interactions with items at a particular point in time). Roughly speaking the contributions are as follows:
(a) the paper models the co-evolutionary process of users' preferences toward items
(b) the paper is able to incorporate external sources of information, such as user and item features
(c) the process proposed is generative, so is able to estimate specific time-points at which events occur
(d) the model is able to account for non-linearities in the above

Following the pre-review questions, I understand that it is the combination of (a) and (c) that is the most novel aspect of the paper. A fully generative process which can be sampled is certainly nice (though of course, non-generative processes like regular old regression can estimate specific time points and such too, so not sure in practice how relevant this distinction is).

Other than that the above parts have all appeared in some combination in previous work, though the combination of parts here certainly passes the novelty bar.

I hadn't quite followed the issue mentioned in the pre-review discussion that the model requires multiple interactions per userXitem pair in order to fit the model (e.g. a user interacts with the same business multiple times). This is a slightly unusual setting compared to most temporal recommender systems work. I question to some extent whether this problem setting isn't a bit restrictive. That being said I take the point about why the authors had to subsample the Yelp data, but keeping only users with "hundreds" of events means that you're left with a very biased sample of the user base.

Other than the above issues, the paper is technically nice, and the experiments include strong baselines and reports good performance.

[Official Review · AnonReviewer3 · rating 6 · confidence 4 · 16 Dec 2016 (modified: 17 Dec 2016)]
**review for Recurrent Coevolutionary Feature Embedding Processes for Recommendation**

This paper proposes a method to model time changing dynamics in collaborative filtering.
Comments:
1) The main idea of the paper is build upon similar to a previous work by the same group of author (Wang et.al KDD), the major difference appears to be change some of the latent factors to be RNN
2) The author describes a BPTT technique to train the model  
3) The author introduced time prediction as a new metric to evaluate the effectiveness of time dependent model. However, this need to be condition on a given user-item pair.
4) It would be interesting to consider other metrics, for example
- The switching time where a user changes his/her to another item 
- Jointly predict the next item and switching time. 
In summary, this is a paper that improves over an existing work on time dynamics model in recommender system. The time prediction metric is interesting and opens up interesting discussion on how we should evaluate recommender systems when time is involved (see also comments).

[Official Review · AnonReviewer2 · rating 6 · confidence 4 · 19 Dec 2016]
**No Title**

The paper introduces a time dependent recommender system based on point processes parametrized by time dependent user and item latent representations. The later are modeled as coupled – autoregressive processes – i.e. the representation of a user/item changes when he interacts with an item/user, and is a function of both the user and the item representations before time t. This is called coevolution here and the autoregressive process is called recurrent NN. The model may also incorporate heterogeneous inputs. Experiments are performed on several datasets, and the model is compared with different baselines.
There are several contributions in the paper: 1) modeling recommendation via parametrized point processes where the parameter dynamics are modeled by latent user/item representations, 2) an optimization algorithm for maximizing the likelihood of this process, with different technical tricks that seem to break its intrinsic complexity, 3) evaluation experiments for time dependent recommendation.
The paper by the same authors (NIPS 2016) describes a similar model of continuous time coevolution, and a similar evaluation. The difference lies in the details of the model: the point process model is not the same and of the latent factor dynamic model is slightly different, but the modeling approach and the arguments are exactly the same. By the end, one does not know what makes this model perform better than the one proposed in NIPS, is it the choice for the process, the new parametrization? Both are quite similar. There is no justification on the choice of the specific form of the point process in the two papers. Did the authors tried other forms as well? The same remark applies for the form of the dynamical process: the non-linearity used for the modeling of the latent user/item vectors here is limited to a sigmoid function, which probably does not change much w.r.t. a linear model, but there is no evidence of the role of this non linearity in the paper. Note that there are some inconsistencies between the results in the two papers.
Concerning the evaluation, the authors introduce two criteria. I did not get exactly how they evaluate the item recommendation: it is mentioned that at each time t, the model predicts the item the user will interact with. Do you mean, the next item the user will interact with after time t? For the time prediction, why is it a relevant metric for recommendation?
A comparison of the complexity, or execution time of the different methods would be helpful. The complexity of your method is apparently proportional to #items*#users, what are the complexity limits of your methods.
Overall, the paper is quite nice and looks technically sound, albeit many details are missing. On the other hand, I have a mixed feeling because of the similarity with NIPS paper. The authors should have make a better work at convincing us that this is not a marginal extension of previous work by the authors.  I was not convinced either by the evaluation criteria and there is no evidence that the model can be used for large datasets.

[Author Response · Hanjun Dai · 13 Jan 2017]
**Paper Revision**

Dear reviewers, we have revised our paper according to your insightful suggestions and comments.

1) We highlight the importance and difficulty of modeling the nonlinearity in the point process models in the introduction part. 

2) We added the discussion with Chen et.al ICML 2013 and a detailed comparison with Wang et.al NIPS 2016.
 
3) We added the experiment on the large Yelp dataset, which contains 1,005 users and 47,924 items. We run our algorithm and all the baselines on this one. The result of our algorithm is consistently better than alternatives. 

4) We added the section 6.4.1 explaining the quantitative results we get in the experiment. Specifically, we studied the performance of different history length of users, and how the diversity of tastes of users affect the results by visualizing the user-item interaction graph. 

5) We also added the section 6.4.2 that quantitatively compare the effect of different history length on Yelp dataset.

[Final Decision · Program Chairs · 06 Feb 2017]
**ICLR committee final decision**

A nice paper, with sufficient experimental validation, and the idea of incorporating a form of change point detection is good. However, the technical contribution relative to the NIPS paper by the same authors is not significant, in that it primarily involves using an RNN instead of a Hawkes process to model the temporal dynamics. The results are significantly better than this earlier paper -- the authors should explore if this is due only to the RNN, or to the optimization method.